# The Endoscopic Retrograde Cholangiopancreatography and Endoscopic Ultrasound Connection: Unity Is Strength, or the Endoscopic Ultrasonography Retrograde Cholangiopancreatography Concept

**DOI:** 10.3390/diagnostics13203265

**Published:** 2023-10-20

**Authors:** Claudio Giovanni De Angelis, Eleonora Dall’Amico, Maria Teresa Staiano, Marcantonio Gesualdo, Mauro Bruno, Silvia Gaia, Marco Sacco, Federica Fimiano, Anna Mauriello, Simone Dibitetto, Chiara Canalis, Rosa Claudia Stasio, Alessandro Caneglias, Federica Mediati, Rodolfo Rocca

**Affiliations:** 1Gastroenterology and Digestive Endoscopy Unit, AOU Città della Salute e della Scienza, University of Turin, 10126 Turin, Italy; marcantoniogesualdo@gmail.com (M.G.);; 2Gastroenterology Department, Mauriziano Hospital, 10128 Turin, Italy

**Keywords:** endoscopic ultrasound, ERCP, biliary stones, biliary stenosis, EDUS, IDUS, EURCP

## Abstract

Endoscopic ultrasound (EUS) and endoscopic retrograde cholangiopancreatography (ERCP) are both crucial for the endoscopic management of biliopancreatic diseases: the combination of their diagnostic and therapeutic potential is useful in many clinical scenarios, such as indeterminate biliary stenosis, biliary stones, chronic pancreatitis and biliary and pancreatic malignancies. This natural and evident convergence between EUS and ERCP, which by 2006 we were calling the “Endoscopic ultrasonography retrograde colangiopancreatography (EURCP) concept”, has become a hot topic in the last years, together with the implementation of the therapeutic possibilities of EUS (from EUS-guided necrosectomy to gastro-entero anastomoses) and with the return of ERCP to its original diagnostic purpose thanks to ancillary techniques (extraductal ultrasound (EDUS), intraductal ultrasound (IDUS), cholangiopancreatoscopy with biopsies and probe-based confocal laser endomicroscopy (pCLE)). In this literary review, we retraced the recent history of EUS and ERCP, reported examples of the clinical applicability of the EURCP concept and explored the option of performing the two procedures in only one endoscopic session, with its positive implications for the patient, the endoscopist and the health care system. In the last few years, we also evaluated the possibility of combining EUS and ERCP into a single endoscopic instrument in a single step, but certain obstacles surrounding this approach remain.

## 1. Introduction

Endoscopic retrograde cholangiopancreatography (ERCP) and endoscopic ultrasound (EUS) were born, respectively, at the end of the 1960s and at the beginning of the 1980s, for the common purpose of achieving accurate instrumental images of the bilio-pancreatic district. Indeed, abdominal ultrasound and computed tomography (CT) scanning at that time were at the very beginning of their history and had poor diagnostic performance.

Despite the diagnostic vocation of its birth, ERCP has been confined to an almost exclusively therapeutic role for a long time (from the beginning to the 2010s) because of the development of other non-invasive, but equally accurate, imaging techniques such as magnetic resonance cholangiopancreatography (MRCP) and EUS. Instead, in the last decade there has been a progressive return to its original diagnostic purpose thanks to the growth and the implementation of so-called “ancillary techniques”, such as IDUS (intraductal ultrasound), cholangioscopy and probe-based confocal laser endomicroscopy (pCLE), finding their main application in the field of indeterminate biliary and pancreatic strictures [1].

Similarly, EUS, which was first developed as an exclusively diagnostic technique, has acquired, in recent years, an increasingly interventional and therapeutic role. This was primarily intended for tissue acquisition (TA) but now allows many EUS-guided therapeutic interventions (drainage of pancreatic fluid collections and WONs, endoscopic necrosectomy, biliary drainage and gastro- and entero-enteric anastomosis). These changes are making EUS a more attractive technique, even among ERCP practitioners, due to the need for and goal of endoscopically managing the heterogeneity in bilio-pancreatic pathologies [2], offering a less-invasive alternative to surgery in appropriate clinical settings.

ERCP and EUS, therefore, developed independently for some years, but at the beginning of the 2000s, and more widely after the 2010s, they started to converge with each other with the implementation of the diagnostic potential of ERCP and the therapeutic potential of EUS. This integration between EUS and ERCP, which we called the “EURCP (Endoscopic Ultrasound Retrograde Cholangiopancreatography) concept”, can be expressed in different configurations and in different clinical scenarios [3].

## 2. How the Convergence between EUS and ERCP Takes Place in Clinical Practice

### 2.1. EUS and ERCP: Necessarily Separate Sessions?

EUS and ERCP are often performed and scheduled in separate endoscopic sessions, occupying different rooms with different equipment and on different days, with the need for repeated sedation, and often changing the instruments and the operator, depending on available expertise.

With proper organisation, it could be possible to manage EUS and ERCP in a single endoscopic session; that would mean reducing the risk of double sedation for the patient, reducing the cumulative time of hospital stay (post-procedural monitoring once instead of twice) and using only one endoscopic room. In a room equipped with both fluoroscopy and EUS equipment, we could first perform an EUS with or without fine-needle aspiration/biopsy (FNA/B) and then an ERCP with or without sphincterotomy (STE), changing only the endoscopic instrument and, if necessary, the operator. Similarly, an EUS could follow an ERCP, if useful, without the need to stop sedation [4].

Some studies have investigated the outcome of a single-session and single-sedation procedure performed with the exchange of instruments compared with two separate endoscopic manoeuvres. The results prove that there is no significant difference in diagnostic and therapeutic success in total procedure time and in adverse event rate. Instead, the dose of propofol administered was significantly lower, as was the duration of hospital stay, providing an accurate diagnosis with less discomfort and improving the quality of life of the patients and the overall cost for “EURCP” procedures [5,6].

### 2.2. Extraductal Ultrasound (EDUS)

EDUS is an endoscopic technique that consists of performing an EUS using a miniaturised ultrasound probe inserted into the working channel of a duodenoscope. In patients with intermediate risk of extra-hepatic biliary lithiasis, the diagnostic performance of EDUS for distal choledochal stones (sensitivity of 90%, specificity of 98% and accuracy of 91%) is very similar to that of EUS (sensitivity of 92%, specificity of 100% and accuracy of 95%) [7]. If EDUS shows the presence of biliary stones, it is possible to proceed to an ERCP with the same duodenoscope, removing the EDUS probe and inserting the devices needed for cannulation, sphincterotomy and extraction of stones. Conversely, if EDUS does not find stones, a useless papilla cannulation with its correlated risks can be avoided.

The main limitation of this method is the poor penetrative power of the high-frequency ultrasound miniprobe (12–20 MHz), confining the best imaging resolution to within 15–20 mm from the duodenal lumen. Therefore, this technique does not allow a panoramic study of the biliary tract and of the pancreas but only of their most distal portions (papilla and terminal tract of the bile duct). In addition, handling a miniprobe can sometimes be challenging, especially using a video duodenoscope [8].

### 2.3. IDUS

Intraductal ultrasound (IDUS) uses a thin calibre miniaturised US probe that is conducted with or without a guide wire through biliary or pancreatic ducts to obtain ultrasound imaging of their lumen, walls and surrounding structures. Similar to EDUS, higher ultrasonographic frequencies confer high resolution but limited penetration depth (29 mm and 18 mm with 12 MHz and 20 MHz probes, respectively) (Figure 1). Remarkably, it does not require a previous endoscopic STE before being performed, and thanks to the wire-guided technology, it is easy to handle and allows a double passage in a biliary or pancreatic duct in 5 to 10 min [9,10].

The clinical scenarios that better show the diagnostic capability of IDUS are choledocholithiasis and indeterminate biliary strictures.

In 2003, Catanzaro and colleagues showed that IDUS could change the endoscopic management in 37% of cases of unclear finding at cholangiography, avoiding biliary cannulation and correlated risks in the absence of stones, or, conversely, identifying stones or sludge and giving a more proper indication of STE [11]. These results sprouted a debate, started by Haber [12], about the utility, in clinical practice, of performing IDUS when ERCP is available, given that cholangiography is the first step of whatever ERCP procedure. Finally, some specific settings where IDUS can overcome the diagnostic limits of cholangiography were defined. A retrospective study suggests that biliary stones smaller than 8 mm and located in a large common bile duct (>12 mm) are the most likely to be missed with ERCP but detected with IDUS. IDUS is also useful in distinguishing stones (echogenic foci with acoustic shadowing) from air bubbles (echoic foci with reverberation artefacts) and biliary sludge (echogenic foci without acoustic shadowing) [13].

Patients with negative or indeterminate ERCP findings who are at high risk of having biliary stones should be evaluated with IDUS: in a prospective comparative study [14], this technique showed an excellent diagnostic yield for bile duct stones, with a sensitivity of 95%, which was significantly higher than MRCP and ERCP (respectively, 80% and 90%). Moreover, IDUS plus ERCP can significantly increase the accuracy of ERCP alone in detecting biliary stones or sludge (97% vs. 87%) [15] (Figure 1b). Therefore, some authors have proposed IDUS to verify the correct clearance of a biliary duct after stone extraction, demonstrating a significant reduction in biliary stone recurrence in patients who underwent ERCP plus IDUS compared with ERCP alone (3.4% vs. 13.2%) [16].

According to ESGE (European Society of Gastrointestinal Endoscopy) guidelines on the management of biliary stones, EUS and MRCP should be preferred in the assessment of patients with suspected biliary stones because of their safety, deeper view and less invasiveness [17], while IDUS finds its best application in the above-mentioned scenarios.

Indeterminate biliary strictures (IBS) are clinical entities where IDUS finds its most successful application. Since its beginning in the 2000s, several studies demonstrated the superiority of IDUS over ERCP plus tissue sampling in the differential diagnosis of IBS, with a sensitivity of 90% vs. 48%, a negative predictive value (NPV) of 90% vs. 64% and an accuracy of 92% vs. 73% [18].

Different ultrasonographic features have been correlated with malignancy, such as bile duct wall thickening > 7 mm, a stenosis length > 20 mm, the irregularity in the layering of the biliary wall, an eccentric wall thickening, a hypoechogenic mass with irregular margins, an irregular papillary surface and the presence of round shaped, hypoechoic lymph nodes [19,20] (Figure 1c and Figure 2a).

Several publications contributed to establishing the role of IDUS in this setting. In 2013, Meister et al. [21] retrospectively analysed the diagnostic performance of ERCP plus IDUS in 397 patients, reporting a sensitivity, specificity and accuracy of 93.2%, 89.5% and 91.4% respectively. In the retrospective study by Heinzow and colleagues comparing IDUS, trans papillary forceps biopsies (ETP), EUS and CT, IDUS showed a malignancy detection rate of 91%, which was significantly higher than ETP (59%), EUS (74%) and CT (73%, *p* < 0.0001 [22]. Also, in our personal experience with 54 patients, IDUS showed a diagnostic accuracy of 86% in the differential diagnosis between malignant and benign biliary stenoses [23]. Despite these brilliant results, the risk of adverse events related to ERCP and IDUS should not be overlooked, especially if compared with the safety of EUS and CT, since IDUS has been described as an independent risk factor for post-ERCP pancreatitis in several studies [22,24,25].

IDUS also shows superiority over EUS in the loco-regional T-staging of malignant lesions of the common bile duct, with an accuracy of 77.7% vs. 54.1%. In contrast, lymph node involvement is underestimated due to the poor penetration depth of the ultrasound miniprobe, with an accuracy of 33.3% [26].

IDUS-guided tissue sampling could represent an interesting future development. Kim and colleagues [27] demonstrated that IDUS-guided trans-papillary biopsies have higher sensitivity than cholangiography-guided biopsies (90.8% vs. 76.9%, *p* = 0.027) in the setting of IBS. This method lacks specific devices combining IDUS and biopsy forceps and currently, the guidelines indicate per-oral cholangioscopy or EUS-guided sampling in indeterminate biliary stenosis [28].

Moreover, IDUS has been demonstrated to be useful in assisting endoscopic trans-papillary gallbladder drainage [29] and in the differential diagnosis between IgG4-related sclerosing cholangitis and primary sclerosing cholangitis or cholangiocarcinoma [30,31]. Lastly, IDUS has been shown to be able to reduce the need for radiocontrast and X-ray exposition in various extrahepatic biliary diseases, which is crucial in particular settings such as ERCP during pregnancy, patients with adverse reactions to iodine-containing contrast media and in critically ill intensive care unit patients necessitating ERCP, in which moving the patient to a setting with facilities for fluoroscopy can represent a substantial obstacle [32].

In 2015, the ESGE technology review about intraductal bilio-pancreatic images [9] concluded that the indications for IDUS were not well-established, but that differential diagnosis of IBS and ampullary tumours could be two promising fields of application. Nowadays, data on IDUS are scant, probably limited by the lack of expertise and availability [33].

### 2.4. EUS and ERCP: Examples of an Integrated Approach

Choledocholithiasis is an important area of applicability of the EURCP concept. ERCP has been considered for decades as the diagnostic and therapeutic gold standard due to the use of ascending cholangiography, allowing an optimal detection of filling defects. However, about 12% of patients undergoing ERCP develop from mild to severe complications such as cholangitis, bleeding, perforation and pancreatitis [34]. Therefore, a diagnostic test that provides good diagnostic performance (a high negative predictive value with an acceptable positive predictive value) and is less invasive and cheaper is needed to overcome the major limits of ERCP. In this perspective, EUS seems to be the technique with the best accuracy to identify lithiasis of the common bile duct, with a sensitivity of 89–94% and a specificity of 94–95% [35], with low risk of adverse events. In a recent study on 131 patients, Pausawasdi et al. claimed that EUS was able to identify bile duct pathologies in 67% of patients with inconclusive MDCT or MRI, with or without MRCP, showing an excellent diagnostic performance for biliary lesions, with an AUROC of 0.98 and a specificity of 98% [36].

However, MRCP also competes with EUS for the role of best diagnostic method for the bilio-pancreatic district. In a recent meta-analysis, Meeralam and colleagues concluded that EUS shows a higher overall sensitivity compared with MRCP (95% vs. 87%), especially for stones below 6 mm, without significant differences in terms of specificity (90% vs. 92%) [37]. Furthermore, considering the cost-effectiveness and cost–utility ratio, different studies suggest that EUS could be more appropriate than MRCP or ERCP as the initial imaging modality in patients with extra-hepatic biliary disease, being similar to ERCP in diagnostic accuracy but with a lower morbidity and mortality rate [38,39]. EUS should therefore be used to avoid unnecessary ERCP and related complications.

Both the American Society for Gastrointestinal Endoscopy (ASGE) and ESGE suggest different clinical management depending on the risk of cholecolithiasis, which is stratified as low (<10%), intermediate (10% to 50%) and high (>50%) risk [17,40]. In 2019, to limit diagnostic ERCP and related risks, the following criteria were defined to identify high-risk patients: the presence of stones in the main biliary duct identified using the US/second-level imaging technique, total bilirubin > 4 mg/dL, dilated main bile duct on imaging (>6 mm with gallbladder in situ or >8 mm in patients with a previous cholecystectomy) or cholangitis. Patients presenting with any of the high-risk criteria should directly undergo ERCP as the first diagnostic and therapeutic approach. On the other hand, patients with intermediate-risk (biochemical test abnormalities, age > 55 years or a dilated bile duct) should be evaluated with EUS or MRCP to assess the real need for ERCP as a therapeutic manoeuvre. In fact, in this setting, EUS seems to avoid 67% of unnecessary ERCPs [41].

In the last few years, however, the hypothesis that EUS may also be indicated in high-risk patients is becoming increasingly stronger. Thanks to its accuracy in detecting and describing biliary lithiasis (number and size of stones), it can be useful in guiding the best therapeutic strategy [42].

Moreover, performing EUS and ERCP in the same session could be advantageous for patients at medium and high risk of CBD lithiasis by obtaining real-time information from EUS that can further lead the endoscopic management, exploiting single-sedation for both diagnosis and treatment, reducing the risk of cholangitis or pancreatitis while waiting for ERCP to be performed and, finally, reducing hospitalisation and costs [41].

IBS can also benefit from this integrated approach. Nowadays, it is possible to offer the patient the so-called “all in one technique”, an integrated endoscopic approach that consists of performing, during the same ERCP session, IDUS, cholangioscopy, pCLE, intraductal tissue acquisition with dedicated forceps, brushing and drainage of the bile duct (Figure 2). This modality maximizes the diagnostic yield, accelerates the diagnosis and reduces the need to repeat one or more ERCPs with related risks [43].

Malignant diseases of the pancreas and biliary ducts are also clinical settings where both EUS and ERCP are often required. For instance, in the case of jaundice due to a pancreatic head lesion, we can obtain a cytologic and histologic sample and a loco-regional staging with EUS while solving the obstruction with biliary duct stenting with ERCP. In the case of chronic pancreatitis (CP), ERCP allows the therapeutic management of pancreatic duct alteration and a diagnostic evaluation if a differential diagnosis between benign and malignant main pancreatic duct (MPD) stricture or dilatation is needed, while EUS provides accurate information about the presence of CP and its severity and, in case of pain, guiding celiac plexus neurolysis. Fluid collections after acute pancreatitis can also be managed with either ERCP (MPD stenting) or EUS (EUS-guided drainage) [44].

Therefore, the clinical settings where an integrated EUS-ERCP approach seems to be the best choice are various and common in clinical practice.

### 2.5. The Last Frontier: Integration Overcoming Limits

Nowadays, multiple therapeutic applications of EUS have been developed to be complementary to ERCP or to overcome some ERCP technical limits with promising results.

Biliary and pancreatic EUS-guided rendez-vous are an example of EUS acting to “pave the way” to ERCP: when trans-papillary access is not feasible, EUS-guided access to the biliary or pancreatic duct is created to allow an anterograde guidewire insertion. The guidewire can be later used for a retrograde manoeuvre with the duodenoscope, allowing an endoscopic approach, even in the case of fibrotic stenosis, impassable stones due to chronic pancreatitis or in the case of post-surgical altered anatomies [45]. The success rate ranges from 70% to 100% for biliary and 70% to 90% for pancreatic rendez-vous; however, the rate of adverse events is increased compared with standard ERCP (10% and 20%, respectively) [46].

The introduction of lumen apposing metal stent (LAMS) [47] with rigid electrocautery-enhanced tips has importantly broadened the therapeutic applications of EUS, with the possibility to perform EUS-guided gastrointestinal anastomosis (EUS-GIA) and their more specific declinations, EUS-guided biliary drainage (EUS-BD), EUS-guided gallbladder drainage (EUS-GBD), EUS-directed trans-gastric ERCP (EDGE) and EUS-guided afferent loop drainage (EUS-A). Growing evidence is available on the efficacy and safety of those endoscopic treatments, which has prompted ESGE to publish dedicated guidelines [48].

In particular, EUS-directed trans-gastric ERCP is a clear example of the “EURCP” concept: the EUS-guided placed LAMS opens a route for the duodenum in patients who underwent a gastric by-pass with a Roux and Y reconstruction. It showed similar outcomes to laparoscopic-assisted ERCP in terms of success rate (96.5% vs. 100%) and adverse event rates (24% vs. 19%), with a substantial reduction in procedural time and length of hospital stay [49]. A recent meta-analysis showed a superiority of the EUS-guided and laparoscopic-assisted approaches compared with enteroscopic-assisted ERCP in terms of stone clearance and procedural time [50].

There are some technical challenges, such as the localization of the targeted loop and the prevention of LAMS dislocation. In addition, the potential adverse events can be extremely complex to handle. Therefore, therapeutic EUS should be considered in the contest of referral centres with experienced endoscopists.

Moreover, there is still debate about post-procedural management, in particular, for the removal time of LAMS and the need for endoscopic follow-up, due to the lack of long-term data. As long as they represent an alternative to surgery with promising outcomes and a wide range of applications, the acquisition of excellent skills both in EUS and ERCP should be strongly encouraged among young endoscopists who are particularly interested in advanced endoscopy [2].

In patients with biliary obstruction, ERCP has always been considered the initial therapeutic approach, with a success rate between 90 and 97% of cases; however, in some situations, such as post-surgical anatomic abnormality, difficult cannulations or inaccessible papillas, it can be ineffective [51]. In similar scenarios, EUS-guided access to the BD is becoming a viable and widely used alternative to classic trans-papillary access (Figure 3). EUS-BD can provide the drainage of both intra- and extra-hepatic ducts, bypassing the stenotic papilla and maintaining an endoscopic, minimally invasive approach. EUS-BD also competes with percutaneous biliary drainage (PTBD), avoiding the need for an external catheter and the related discomfort for patients and possible complications, such as infections, bleeding, dislocation or bile leak. Therefore, according to ESGE guidelines, in the event of ERCP failure, EUS-BD should be preferred instead of PTBD [48].

Several studies tried to answer the question “can EUS-BD be proposed as a primary treatment modality instead of ERCP in presence of malignant distal biliary obstruction (MBO)?” Three randomized controlled trials (RTCs) compared EUS with ERCP as a primary therapeutic modality for biliary drainage in cases of MBO [52,53,54]. Clinical outcomes in terms of clinical success and adverse events (AEs) were equal in the two trials [45,47], whereas EUS-BD was found to be superior in terms of stent patency time, stent dysfunction, adverse events rate and need for re-interventions in the trial of Paik et al. [53].

Two recent meta-analyses by Lou et al. [55], in 2019, and by Kakked et al. [56], in 2020, showed similar technical and clinical success rates between EUS-BD and ERCP in the setting of MBO, with a lower adverse events rate for EUS-BD and a higher rate of stent dysfunction and re-interventions for ERCP.

However, more recent systematic reviews and meta-analyses [57,58,59,60] recommended further high-quality RCTs ahead of retiring ERCP as the first approach for patients with distal MBO.

Some authors have proposed that a combination of ERCP and EUS-BD (CERES) can also find a role in malignant hilar biliary obstruction. Particularly, Bismuth type III and IV cholangiocarcinomas cannot be completely drained with ERCP, requiring a subsequent PTBD with the previously described risks and pitfalls.

Kongkam and colleagues [61] suggested a combined approach by placing a SEMS with ERCP to drain one hepatic lobe, and a EUS-guided hepatico-gastrostomy (EUS-HGS) or hepaticoduodenostomy (EUS-HDS) according to the liver lobe to be drained (left and right lobe, respectively). Technical and clinical success rates are promising (84.2% and 76.5%, respectively), without significant differences with PTBD and a lower recurrent biliary obstruction [62].

Similarly, EUS-guided pancreatic duct drainage (EUS-PDD) has been proposed to overcome pancreatic duct retrograde inaccessibility due to chronic pancreatitis, post-surgical anastomotic strictures or altered ductal system. Mainly, two techniques have been described. Pancreatic rendez-vous-assisted ERCP consists of the puncture of the MPD under EUS view, with the passage of a guidewire through the needle in an anterograde fashion to lead a subsequent ERCP. When this solution also fails, a direct EUS-PDD can be attempted. While the first steps are equal to a pancreatic rendez-vous (identification of the MPD, puncture and passage of the guidewire), in a direct EUS-PDD, a dilation of the transmural tract is performed, and a stent is placed between the MPD and the stomach. Since this procedure shows a higher risk of adverse events and stent dysfunction (up to 55% of cases), more studies and dedicated devices are needed to better define its indication and outcomes [46].

### 2.6. The EURCP Concept: Between the Past and an Integrated Future

Given the multitude of possible applications, over the years, we tried to put into practice the true concept of EURCP, developing a single endoscopic instrument combining the diagnostic role of EUS and the therapeutic role of ERCP: the “EURCP or ERCP-FNA scope”. In Turin, from 2002 to 2004, we studied a mechanical radial echoduodenoscope (JF-UM20; Olympus Optical Co., Tokyo, Japan) equipped with an operating channel of 2.2 mm and a radial US scanner with a single frequency of 7.5 MHz [3]. We evaluated its feasibility in 19 patients at intermediate risk of choledocholithiasis. All patients underwent EUS and ERCP (if required) in the same session with the same instrument. Therapeutic interventions with this tool were needed in 16 out of 19 patients (84%). The presence of biliary sludge or stones suspected with EUS was always confirmed using ERCP, with sphincterotomy and exploration of the bile duct, respectively, in 12 patients and 4 patients. No patient required a subsequent ERCP to treat any residual stones and, above all, there were no short or long-term complications (30 days). Since it was a feasibility study, the enrolling process was extremely selective and represented the main limit of this study. In addition, the echoduodenoscope had some disadvantages, such as the single-ultrasound frequency, the radial scan, the very small operating channel, the 80° field of view and a vision based on optical fibre. Although at that time, we demonstrated the feasibility, safety and usefulness of this single-step single-instrument approach, further technical improvements were certainly needed.

In order to optimize these technical features on the way to a real “EURCP or ERCP-FNA scope”, in 2004, we started a long and complex project in collaboration with Olympus Optical Co., Tokyo, which, in 2016, eventually led to the production of the Olympus prototype JF-Y0006-UC. The instrument subsequently successfully passed the feasibility study on models in Turin and then started a long approval process with the European regulatory agency for use in humans, which successfully ended in 2018 (Figure 4).

Unfortunately, the problems raised by increasingly numerous outbreaks of multidrug-resistant organism infections, sometimes related to the difficult cleaning and disinfection of endoscopes with complex tips (mainly duodenoscopes, but also EUS scopes), prompted the Japanese company first to stop and then definitively cancel the EURCP prototype project, nullifying many years of work just before the start of a feasibility technical study in human.

So far, the advantages of the EURCP technique, theoretically postulated and demonstrated on a small group of patients many years ago, are yet to be demonstrated, but surely, this convergence opens new horizons in the management of bilio-pancreatic diseases, regardless of the technical possibility of developing an “almighty instrument”. So, the EURCP scope dream might be definitively over, but not the EURCP concept.

We firmly believe that implementing the EURCP concept in clinical practice, with adequate tools in expert hands, can bring substantial benefits, primarily to patients but also to the healthcare system in terms of cost management. From this perspective, it becomes essential that the concept of convergence is also translated into an education issue to adequately train future generations of endoscopists with a special interest in bilio-pancreatic endoscopy [2].

## 3. Conclusions

The combination of EUS and ERCP performed in a single session with a single sedation, in the same room and, possibly, by the same endoscopist can improve the diagnostic and therapeutic success of the procedure. EUS can provide useful information about the anatomy of the biliary and pancreatic district, the diagnosis and the staging of the lesion under examination and, last but not least, the right indication to the following ERCP.

The advantage is then represented by the possibility of sparing other imaging modalities, such as CT or MRI, which can sometimes be more complex to manage for patients or for health workers (i.e., requiring mobilization of the patient from the radiology department to the endoscopy unit) or more expensive. Furthermore, an EUS performed just before an ERCP guarantees a sort of “real time” imaging evaluation of the clinical scenario, allowing one to establish the presence of biliary stones, the exact location and ultrasound appearance of a stricture and more generally, making available all the information that EUS can provide with great accuracy.

An EURCP procedure is particularly useful whenever we expect an endoscopic intervention that will require an accurate diagnostic evaluation (also with tissue acquisition if needed) before therapeutic manoeuvres, giving the endoscopist almost all information and any therapeutic technical potential he or she may need in daily practice.

On the other hand, the two procedures performed together entail a longer time of sedation, which cannot be proposed for more frail patients. In some endoscopic units, performing EUS in a radiologic room can represent a waste and sometimes the positioning of the duodenoscope in the second duodenum may be more difficult and less stable after a previous EUS examination. Moreover, the EURCP concept requires good skills both in EUS (diagnostic and therapeutic) and in ERCP to be effective but achieving good expertise in both techniques is demanding and time-consuming, with a steep learning curve.

To conclude, in the era of therapeutic EUS and of the renewed diagnostic role of ERCP, we really think that unity is the power and that being able to handle both techniques should become the goal of every endoscopist interested in a complete and effective management of hepato–pancreato–biliary diseases. The EUS-ERCP convergence or EURCP concept today can represent the integrated future of the 21st century bilio-pancreatic endoscopy.

## Figures and Tables

**Figure 1 diagnostics-13-03265-f001:**
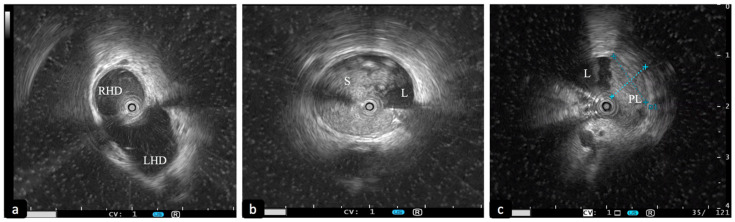
Intraductal ultrasound (IDUS) with 20 MHz probe. (**a**) Slightly dilated biliary confluence. (**b**) Dilation of the CBD with sludge. (**c**) Polypoid intraductal lesion of the CBD (D1: 11 mm; D2: 9 mm). CBD: common bile duct. RHD: right hepatic duct. LHD: left hepatic duct. S: sludge. L: lumen. PL: polypoid lesion.

**Figure 2 diagnostics-13-03265-f002:**
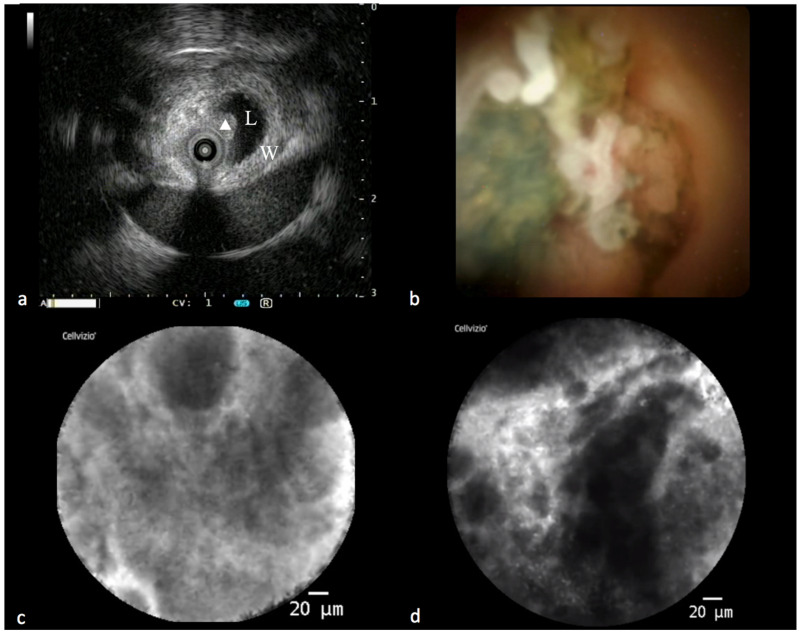
Findings in a malignant biliary stricture during the “all in one technique”. (**a**) IDUS showing eccentric thickening of the bile duct wall with a polypoid profile (L: lumen; W: bile duct wall; triangle: polypoid-shaped thickening). (**b**) Cholangioscopic vision of a polypoid lesion of a common bile duct. (**c**,**d**) PCLE showing some features of malignancy according to the Miami classification for biliary stenosis (thick dark and white bands and dark clumps).

**Figure 3 diagnostics-13-03265-f003:**
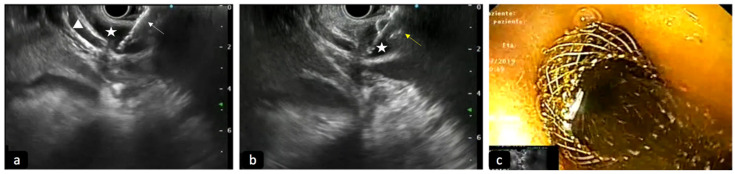
Deployment of LAMS into a dilated common bile duct (CBD) to perform EUS-guided biliary drainage. (**a**) Cystotome of LAMS (white arrow) conducted into the CBD (star) under EUS-guide; triangle: cystic duct. (**b**) EUS view of the opening of the distal flange of LAMS (yellow arrow) into the CBD (star). (**c**) Luminal view of proximal flange opened with bile outflow.

**Figure 4 diagnostics-13-03265-f004:**
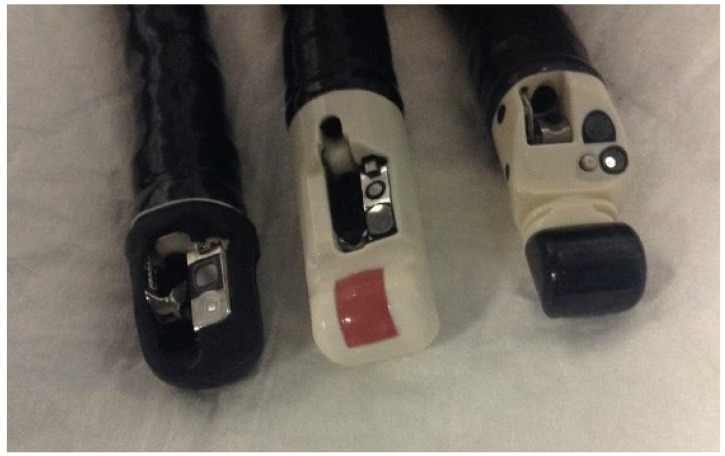
On the left is a standard ERCP scope; on the right is an operative EUS scope; and in the middle, is the “ERCP-FNA” or “EURCP” scope.

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
