# Peer review of "The Endoscopic Retrograde Cholangiopancreatography and Endoscopic Ultrasound Connection: Unity Is Strength, or the Endoscopic Ultrasonography Retrograde Cholangiopancreatography Concept"

_diagnostics, 2023, doi:10.3390/diagnostics13203265_

Round 1

Reviewer 1 Report

The authors review the indications for simultaneous EUS and ERCP and the previous reports on indications, including their experiences. They also presence their experience with ERCP using a radial echoduodenoscope and a prototype endoscope capable of EUS-FNA and ERCP.

This is an interesting report; however, it seems to be required a revision.

Major

In Figure 3, please clearly explain to the reader the location of the bile duct and the EUS-released stent.

Author Response

Thank you very much for your revision.

We reviewed all the images and related explanation, that you can see in the attached file. 

Please let us know if there are any further corrections suggested. 

Reviewer 2 Report

An interesting and practical concept, which has been conveyed in a decent manner

The following are suggested to enhance the manuscript:

1. The potential advantages and disadvantages should be mentioned for this emerging concept

2. The limitations of the combination should be mentioned

3. The available data on this concept should be highlighted

4, The following maybe referenced and discussed:

a. Kongkam P, Tasneem AA, Rerknimitr R. Combination of endoscopic retrograde cholangiopancreatography and endoscopic ultrasonography-guided biliary drainage in malignant hilar biliary obstruction. Dig Endosc. 2019 Apr;31 Suppl 1:50-54. doi: 10.1111/den.13371. PMID: 30994233.

English is largely acceptable

Author Response

Thank you for very much for your useful revision. We answered to your suggestion elaborating a "conclusion" paragraph that you can read below:

Conclusion

The combination of EUS and ERCP performed in a single session with a single sedation, in the same room and, possibly, by the same endoscopist can improve the diagnostic and therapeutic success of the procedure. EUS can provide useful information about the anatomy of biliary and pancreatic district, the diagnosis and the staging of the pathological situation under examination and, last but not least, the right indication to the following ERCP.

The advantage is then represented by the possibility of sparing other imaging modalities, such as CT or MRI, which can sometimes be more complex to manage for the patient (for example difficult to book as outpatients) or for the health workers (requiring a mobilization of the patient from the radiologic department to the endoscopic department) or more expensive. Furthermore, an EUS performed just before the ERCP guarantees a sort of “real time” imaging evaluation of the clinical scenario, allowing to establish with great accuracy the presence of biliary stones or the exact location and ultrasound appearance of a stenosis and more generally making available all the information that EUS can provide.

An EURCP procedure is particularly useful whenever we expect an endoscopic intervention that will require an accurate diagnostic evaluation (also with tissue acquisition if needed) before therapeutic manoeuvres, giving the endoscopist almost all information and any therapeutic technical potential he or she may need in daily practice.

On the other hand, the two procedures performed together entail a longer time of sedation, which can not be proposed to more frail patients. In some endoscopic unit performing EUS in a radiologic room can represent a waste and sometimes the positioning of the duodenoscope in the second duodenum may be more difficult and less stable after a previous EUS examination. Moreover, the EURCP concept requires good skills both in EUS (diagnostic and therapeutic) and in ERCP to be really effective, but achieving a good expertise in both techniques is demanding and time-consuming, with a steep learning curve.

To conclude, in the era of therapeutic EUS and of the renewed diagnostic role of ERCP, we really think that unity is the power and that being able to handle both techniques should become the goal of every endoscopist interested in a complete and effective managing of hepato-pancreato-biliary pathologies. The EUS-ERCP convergence or EURCP concept today can really represent the integrated future of the 21st century bilio-pancreatic endoscopy.

We also dedicated a little paragraph to the reference you suggested:

Some authors have proposed that a combination of ERCP and EUS-BD (CERES) can find a role also in malignant hilar biliary obstruction, in particular for Bismuth III and IV, which can not usually be completely drained by ERCP-placed SEMS and consequently often require PTBD, which leads to a worsening of life-quality for patients and increases infectious risk. Kongkam and colleagues (1) suggest combining the placement of a SEMS with ERCP bypassing the hilary stricture to drain one hepatic lobe, with the EUS-guided placement of a hepato-gastric stent (HGS) in the left lobe or a hepato-duodenal stent (HDS) in the right lobe. If there is atrophy of the right lobe, they recommend draining the left lobe with an EUS-guided hepato-gastrostomy (EUS-HGS); instead, if the lobe to be drained is the right one because of left-lobe atrophy, they recommend performing an EUS-guided hepatico-duodenostomy (EUS-HDS). The technical and clinical success rates are promising (84.2 % and 76.5% respectively, without significant difference with PTBD performance), and some recent data show a lower recurrent biliary obstruction compared with PTBD (2).

  1. Kongkam P, Tasneem AA, Rerknimitr R. Combination of endoscopic retrograde cholangiopancreatography and endoscopic ultrasonography-guided biliary drainage in malignant hilar biliary obstruction. Dig Endosc. 2019;31 Suppl 1:50-54. doi:10.1111/den.13371
  2. Kongkam P, Orprayoon T, Boonmee C, et al. ERCP plus endoscopic ultrasound-guided biliary drainage versus percutaneous transhepatic biliary drainage for malignant hilar biliary obstruction: a multicenter observational open-label study. Endoscopy. 2021;53(1):55-62. doi:10.1055/a-1195-8197

Please see the attachment for the whole work revised.

Please let us know if there are further correction suggested.

Reviewer 3 Report

This is an article that brings a lot of news and is very useful to gastroenterologists. The innovative ideas in the article have a high probability of seeing them put into practice in the near future.

Author Response

Thank you very much for your positive comment to our work.

Reviewer 4 Report

This is an excellent review of the current role and outcomes of combined ERCP and EUS. The paper is well-structured and well-documented. The article highlights the current indications and advantages of performing ERCP and EUS in the same session. Nevertheless, the report is well-illustrated.

Only minor issues:

Please consider introducing a conclusion part highlighting the main ideas of the review.

In line 310, please consider replacing the term “huge meta-analysis”.

Please provide a clearer image of Figure 2b and be more explicit with the other figures (particularly figures 1a, 1b, 2 a, 3).

A few abbreviations are useless since they are not extensively used after that (i.e., MDRO, etc.).

The manuscript should be revised to improve fluency and correct minor spelling, grammar, and editing errors.

Minor editing of English language required

Author Response

Thank you very much for your punctual suggestions. 

For the revised images, please see the attachment.

For the "conclusion" paragraph, you can also read it here:

Conclusion

The combination of EUS and ERCP performed in a single session with a single sedation, in the same room and, possibly, by the same endoscopist can improve the diagnostic and therapeutic success of the procedure. EUS can provide useful information about the anatomy of biliary and pancreatic district, the diagnosis and the staging of the pathological situation under examination and, last but not least, the right indication to the following ERCP.

The advantage is then represented by the possibility of sparing other imaging modalities, such as CT or MRI, which can sometimes be more complex to manage for the patient (for example difficult to book as outpatients) or for the health workers (requiring a mobilization of the patient from the radiologic department to the endoscopic department) or more expensive. Furthermore, an EUS performed just before the ERCP guarantees a sort of “real time” imaging evaluation of the clinical scenario, allowing to establish with great accuracy the presence of biliary stones or the exact location and ultrasound appearance of a stenosis and more generally making available all the information that EUS can provide.

An EURCP procedure is particularly useful whenever we expect an endoscopic intervention that will require an accurate diagnostic evaluation (also with tissue acquisition if needed) before therapeutic manoeuvres, giving the endoscopist almost all information and any therapeutic technical potential he or she may need in daily practice.

On the other hand, the two procedures performed together entail a longer time of sedation, which can not be proposed to more frail patients. In some endoscopic unit performing EUS in a radiologic room can represent a waste and sometimes the positioning of the duodenoscope in the second duodenum may be more difficult and less stable after a previous EUS examination. Moreover, the EURCP concept requires good skills both in EUS (diagnostic and therapeutic) and in ERCP to be really effective, but achieving a good expertise in both techniques is demanding and time-consuming, with a steep learning curve.

To conclude, in the era of therapeutic EUS and of the renewed diagnostic role of ERCP, we really think that unity is the power and that being able to handle both techniques should become the goal of every endoscopist interested in a complete and effective managing of hepato-pancreato-biliary pathologies. The EUS-ERCP convergence or EURCP concept today can really represent the integrated future of the 21st century bilio-pancreatic endoscopy.

Round 2

Reviewer 4 Report

The authors adequately addressed all important concerns raised by the reviewers

Author Response

Thank you very much for your cooperation